# Apoptosis in the Pancreatic Cancer Tumor Microenvironment—The Double-Edged Sword of Cancer-Associated Fibroblasts

**DOI:** 10.3390/cells10071653

**Published:** 2021-07-01

**Authors:** Ester Pfeifer, Joy M. Burchell, Francesco Dazzi, Debashis Sarker, Richard Beatson

**Affiliations:** School of Cancer & Pharmaceutical Sciences, King’s College London, London SE1 9RT, UK; ester.pfeifer@kcl.ac.uk (E.P.); joy.burchell@kcl.ac.uk (J.M.B.); francesco.dazzi@kcl.ac.uk (F.D.); debashis.sarker@kcl.ac.uk (D.S.)

**Keywords:** pancreatic cancer, PDAC, fibroblasts, CAF, stroma, TME, tumor microenvironment, apoptosis

## Abstract

Pancreatic ductal adenocarcinoma (PDAC) is associated with poor prognosis. This is attributed to the disease already being advanced at presentation and having a particularly aggressive tumor biology. The PDAC tumor microenvironment (TME) is characterized by a dense desmoplastic stroma, dominated by cancer-associated fibroblasts (CAF), extracellular matrix (ECM) and immune cells displaying immunosuppressive phenotypes. Due to the advanced stage at diagnosis, the depletion of immune effector cells and lack of actionable genomic targets, the standard treatment is still apoptosis-inducing regimens such as chemotherapy. Paradoxically, it has emerged that the direct induction of apoptosis of cancer cells may fuel oncogenic processes in the TME, including education of CAF and immune cells towards pro-tumorigenic phenotypes. The direct effect of cytotoxic therapies on CAF may also enhance tumorigenesis. With the awareness that CAF are the predominant cell type in PDAC driving tumorigenesis with various tumor supportive functions, efforts have been made to try to target them. However, efforts to target CAF have, to date, shown disappointing results in clinical trials. With the help of sophisticated single cell analyses it is now appreciated that CAF in PDAC are a heterogenous population with both tumor supportive and tumor suppressive functions. Hence, there remains a debate whether targeting CAF in PDAC is a valid therapeutic strategy. In this review we discuss how cytotoxic therapies and the induction of apoptosis in PDAC fuels oncogenesis by the education of surrounding stromal cells, with a particular focus on the potential pro-tumorigenic outcomes arising from targeting CAF. In addition, we explore therapeutic avenues to potentially avoid the oncogenic effects of apoptosis in PDAC CAF.

## 1. Introduction

Pancreatic ductal adenocarcinoma (PDAC) has one of the worst prognoses among solid cancers with a 5-year survival of less than 9% [1], with these numbers predicated to worsen [2]. The high mortality is mainly attributed to the particularly aggressive biology of PDAC, coupled with the late presentation and diagnosis at advanced stages when curative surgery is not feasible [3]. Despite some improvements in the management of PDAC, there are still limited molecular-targeted therapies available, hence palliative cytotoxic therapies have remained the standard of care for PDAC with predominately modest efficacy [4]. PDAC is largely resistant to conventional drug therapy and this is partially attributed to its complex and heterogenous tumor microenvironment (TME), with typically about 85% of the tumor being composed of a very dense, hypoxic, desmoplastic stroma depleted of immune effector cells upon diagnosis [5]. Cancer-associated fibroblasts (CAF) are the main cellular component in PDAC and once activated by injury or chronic inflammation they deposit large amounts of extracellular matrix (ECM), including laminins, fibronectins, collagens and hyaluronan, as well as other factors such as TGFβ1 and prostaglandin E2 (PGE_2_) [6]. Prominent desmoplasia is an indicator of poor prognosis and treatment resistance [7]. Not only is it difficult for drugs to reach tumor cells surrounded by a dense desmoplastic stroma, but it is well established that the CAF promote tumor progression by enhancing desmoplasia, inducing treatment resistance, and promoting tumor growth and metastasis [8,9,10,11,12]. Increased understanding of the role of CAF in driving tumor progression identified them as targets; as a treatment option alone or in combination with standard therapies. However, different CAF-targeting strategies have to date yielded no clear evidence of improved survival in patients with advanced PDAC; indeed, some studies have suggested inferior outcomes compared to best standard of care [13]. These disappointing findings raised the question whether CAF could have both tumor repressive and tumor-promoting functions. With the use of sophisticated mouse models, single cell analysis, spatial mapping platforms, pathological associations and ablation models, there is now greater insight into the diverse CAF populations present in the TME. As such we now know that PDAC CAF are heterogenous with factors such as origin, cell surface marker expression, cytokine production, cell signaling, cell–cell interactions and gene expression, playing a role in determination of their biological behavior and clinical impact [14].

Originally, it was believed that PDAC CAF were composed of a homogeneous population with regards to their morphology, location and origin (being derived from resident pancreatic stellate cells [PSCs]) [15]. It is now known that CAF in PDAC are a mixture of stromal cells from various sources [16]. Under normal physiological conditions, only 4–7% of the cells in the pancreas represent PSC; however, upon activation by pancreatic injury (such as carcinogenesis), PSCs differentiate into a myofibroblast-like phenotype, becoming CAF [17,18]. Another well-known source of CAF are mesenchymal stromal cells (MSCs) which have a high tropism to migrate to sites of injury and, once in the TME, can differentiate into CAF. Fibrocytes and mesothelial cells can also be recruited and activated to become CAF once in the TME. Epithelial and endothelial cells within the pancreas can undergo epithelial and endothelial to mesenchymal transition (EMT and EndoMT) and exhibit a fibroblastic phenotype [19,20]. Other minority groups of cells believed to be educated to CAF-like cells include adipocytes and pericytes [21].

One of the greatest challenges in studying subpopulations of CAF in the PDAC TME is a lack of CAF-specific markers; the CAF population has previously been identified by an absence of markers to define other cell types: EPCAM for epithelial cells, CD45 to mark leukocytes and CD31 to mark endothelial cells. However, several markers have been identified to be expressed on CAF, including platelet-derived growth factor receptor-β (PDGFR-β/*PDGFRB*), alpha smooth muscle actin (α-SMA/*ACTA2*), fibroblast activating protein (*FAP*), fibroblast-specific protein 1 (FSP1/*S100A4*), podoplanin (*PDPN*) and Meflin (*ISLR*) [14,22]. However, the expression of these markers is not uniform, with variability among different CAF populations [11]. A further obstacle in translating CAF-targeting strategies using these markers in vivo is that many of the CAF markers also are expressed on other cell types in other organs. For instance, αSMA is found on pericytes throughout the body, FAP on mesenchymal cells in bone and muscle and PDPN [23] is also expressed on endothelial cells in the lymphatic system [14].

The interactions between CAF and their microenvironment are highly complex. As mentioned, evidence is emerging that CAF have different functions in the TME ranging from tumor promoting to tumor suppressing, with some maintaining important homeostatic roles. For instance, there are studies that have demonstrated that the presence of certain sub-types of CAF are associated with more aggressive tumors and shorter survival [24]. Initial studies trying to differentiate the various CAF functions within the PDAC stroma identified two distinct subpopulations of CAF: a) myofibroblastic CAF (myCAF) expressing high levels of α-SMA that secrete trophic factors and extracellular matrix components to construct the extracellular matrix, and b) inflammatory CAF (iCAF) exhibiting low expression of α-SMA and high expression of inflammatory mediators, such as interleukin-6 (IL-6), IL-11 and leukemia inhibitory factor (LIF) through IL-1 induced JAK/STAT activation [11,25,26].

MyCAF and iCAF are two examples of the heterogeneous CAF populations present in PDAC. Cancer cells drive this transcriptional heterogeneity by reprogramming CAF of different origins which then respond to the complex signaling in the TME in different manners. The growing interest in understanding the different populations of CAF in the PDAC TME has led to many recent studies carried out at single cell resolution [27,28,29,30]. Further details of the complex composition of CAF populations in PDAC will not be discussed here, as this has been extensively reviewed recently [10,11,14,31,32,33].

Despite increasing knowledge about the heterogeneity of CAF and their importance in both normal tissue homeostasis and in tumorigenesis, they have often been neglected when developing and administering anti-cancer therapies. There is limited understanding of how CAF (especially different sub-populations of CAF) are affected by conventional anti-cancer therapies, both directly and indirectly. This review will focus on the often-neglected oncogenic processes arising by cell death in the TME and how apoptosis and administration of cytotoxic therapies in PDAC potentially fuel oncogenesis by the education of CAF into tumor-promoting phenotypes. We will then review the potential pro-tumorigenic outcomes arising from the ablation of CAF in PDAC before touching on novel targeting strategies that could avoid the oncogenic effects of direct apoptosis and depletion in PDAC CAF.

## 2. Apoptosis Fueling Tumorigenesis

Under normal physiological conditions, somatic cells undergo mitosis and, after a certain number of divisions, apoptosis. When this balance is disturbed, disease, including cancer can develop, either by an increase in cell proliferation or a decrease in cell death. Apoptosis in cancer cells has largely been associated as a positive phenomenon, with the main goal of cancer treatment being to induce maximal death in cancer cells and minimal damage to healthy cells. However, even though apoptosis is a cell-autonomous event, when put into a complex microenvironment, the effects of apoptosis are not uniformly positive and paradoxically, many of these effects have been seen to fuel oncogenesis. As a matter of fact, many studies in different cancer types have shown that higher levels of apoptosis is correlated with poorer outcomes [34,35,36,37,38,39].

It is known that apoptotic cells can promote proliferation in surrounding cells, a process also known as apoptosis-induced proliferation (AiP). This compensatory mechanism was first observed in *Drosophila melanogaster*, where caspase-dependent apoptosis in the imaginal disk was shown to activate mitogenic signaling, promoting the proliferation of neighboring cells [40,41,42]. In tumorigenesis, when cytotoxic agents are administered, compensatory proliferation may be sustained for longer periods of time, which may allow repopulation and proliferation of more aggressive and treatment-resistant clones. In mammals, PGE_2_ has shown to be a key mediator in AiP. When caspases are cleaved, calcium-independent phospholipase A2 (iPLA2) is activated, resulting in an increased production of arachidonic acid which is converted, via cyclooxygenase 1 (COX1) or cyclooxygenase 2 (COX2), to PGE_2_. PGE_2_ in the context of cancer has been shown to promote stem and progenitor cell proliferation and angiogenesis, inhibit T-cell activation and educate immune cells to tumor-promoting phenotypes [43,44]. The notion of PGE_2_ being released by tumor cells following conventional cytotoxic treatments has led to efforts in trying to target PGE_2_, which was shown to be effective by the inhibition of cancer stem cell (CSC) repopulation, reduce incidence and mortality and enhance the effectiveness of chemotherapeutic regimens [45,46,47].

The modulation of immune cells in the TME by apoptotic cells is often neglected in pre-clinical studies. Anti-cancer therapies have historically been tested using human tumors xenotransplanted into immunocompromised mice [48,49] where the role of the immune system cannot be analyzed. Apoptotic cells, as part of a microenvironment, induce an immunogenic process as they release so-called “eat me” signals enabling effective clearance by phagocytosis [50]. Molecules such as fractalkine, phosphatidylserine (PS), lysophosphatidylcholine and sphingosine-1-phosphate and nucleotides ATP and UTP are released from apoptotic cancer cells and are known to activate or inhibit immune cells and promote efferocytosis [51]. With regards to apoptotic cell clearance in the TME, it has emerged that phagocytosis has a role in initiation, progression and metastasis [52]. Apoptotic cell clearance via efferocytosis polarizes tumor-associated macrophages (TAM) to adopt pro-tumor ‘M2’ wound-healing phenotypes which, by secreting cytokines such as IL-4, IL-10, IL-13 and TGF-β1, further educate Th0 or Th1 CD4 T cells towards a Th2 phenotype [41]. Efferocytosis can also recruit Tregs, further suppressing effector T cells [53]. These changes result in ineffective or severely compromised anti-cancer immune responses. Further details of the oncogenic processes of apoptotic cancer cell immune responses including efferocytosis can be found in recent reviews [41,52,54,55].

Apoptotic cells also release extracellular vesicles (EVs) as a way to communicate with proximal and distal sites. EVs released from apoptotic cells are membrane-delimited vesicles that display a broad size heterogeneity ranging from exosomes (30–150 nm) and microvesicles (100–1000 nm) to larger apoptotic blebs/bodies (50 to 5000 nm) [56]. These vesicles carry cargos containing proteins, nuclear and cytosolic DNA and RNAs (e.g., intact and fragmented messenger RNA, microRNA, transfer-RNA and ribosomal RNA fragments), metabolites including amino acids and lipids to stimulate proliferation and recruit cells to the TME. The exact function of EVs released from apoptotic cells in human cancers remains unclear, but growing evidence suggests that they may play a role in oncogenesis [57]. Apoptotic bodies can carry organelles and nuclear components such as DNA, including oncogenes, that can be delivered to neighboring cells, potentially having a tumorigenic function [55]. Apoptotic EVs from glioblastoma cells have been shown to carry spliceosomes, which in recipient cells promoted proliferation and resistance to therapy [58]. Further, it has been documented that EVs released from squamous head and neck cancer cells promote migration in the recipient cancer cells [59]. EVs from apoptotic cells also aid in the effective clearance and efferocytosis of apoptotic cells by exposing factors such as phosphatidylserine on their surface for effective recognition and removal by phagocytic cells. Apoptotic cells have also been shown to release EVs carrying fractalkine, known to induce chemotaxis and efferocytosis [60]. In addition, fractalkine has been shown to have other pro-tumorigenic functions, for instance, enhancing cancer cell migration, metastasis and angiogenesis [61]. Many other immunostimulatory and immunosuppressive effects of apoptotic EVs have been reported [62,63,64,65], altogether implying that the release of apoptotic EVs has an important role in the modulation of the TME and in oncogenesis.

Recently, there have been a number of reviews published giving further details about the oncogenic effects of apoptosis in cancer cells [41,52,55,66]; however, information about the direct effects on CAF is sparse. The following sections will discuss how cell death in the TME educate CAF to drive pro-tumorigenic processes.

## 3. The Education of CAF into a Pro-Tumorigenic Phenotype by Dying Cancer Cells

Paradoxically, apoptosis induction in cancer cells can cause undesired effects and even promote tumor progression. Initial studies, performed in *Drosophila,* observed compensatory proliferation occurring in imaginal disc compartments neighboring those in which apoptosis was induced [67]. These processes were regulated by conserved pathways, such as Wnt-β-catmorphogenetics bone morphogenetic protein (BMP) and hedgehog (Hh) [68], and have also been reported in mammalian tissues. Hh signaling, for example, has been shown to link the apoptosis of hepatocytes with compensatory proliferation of liver progenitors and myofibroblasts [69]; hence, one can hypothesize that apoptotic cancer cells could induce compensatory proliferation not only in other cancer cells, but potentially in other surrounding cell types such as stromal cells.

Apoptosis and necrosis are the two major types of tumor therapy-induced cell death [70]. Apoptosis includes ‘extrinsic’ and ‘intrinsic’ pathways and both result in characteristic cellular changes, including cell shrinkage and fragmentation into membrane-bound apoptotic bodies followed by immunological activation and phagocytosis [71]. If the apoptotic cell is not recognized and cleared, this may be succeeded by secondary necrosis, which is characterized by disruption of the cellular membrane, swelling of the cytoplasm and mitochondria and breakdown of organelles. As cells lose their cell membrane integrity, induction of acute inflammation and activation of innate immune cells occur via several factors such as cytokines, chemokines and damage-associated molecular patterns (DAMPs). Most anticancer therapies promote DAMP release [70]. Gemcitabine, one of the most frequently used cytotoxics for treatment of PDAC, was shown to trigger DAMP release (e.g., calreticulin, HSP70, and HMGB1); however, it was unable to induce immunogenic cell death. Further, it was shown that gemcitabine triggered PGE_2_ release as an inhibitory DAMP to counterpoise the adjuvanticity of immunostimulatory DAMPs [72]. These data suggest that gemcitabine administration can potentially fuel oncogenesis by directly releasing PGE_2_ and inducing AiP, in addition to suppressing the immune response in an already immunosuppressive TME.

DAMPs released by damaged tissue or dying cancer cells have also been shown to activate CAF [73]. For instance, fibroblasts in mouse and human breast tumors were able to sense DAMPs, which in turn activated the NLRP3 inflammasome pathway and secretion of IL-1β. The NLRP3 (NOD-, LRR- and pyrin domain-containing protein 3) inflammasome pathway enables cells to react to microbial and damage-associated motifs through the caspase-1 dependent release of factors such as IL-1β and IL-18. This CAF-derived inflammasome signaling enhanced tumor growth and metastasis by modulating the TME towards an immune suppressive milieu and by upregulating the expression of adhesion molecules on endothelial cells [74].

HMGB1 is the DAMP most associated with cancer and has been shown to play an active role in cancer-associated inflammation and pathogenesis. HMGB1 is a potent TLR activator, and the link between DAMP induced TLR activation, chronic inflammation and carcinogenesis has been demonstrated in previous studies [75,76]. In PDAC it has been shown that TLR7 ligation drives stromal inflammation and results in loss of both PTEN and p16 and increased expression of p21, p27, p-p27, p53, Rb, c-Myc, TGF-β1 and SHPTP1. Further, inhibiting, deleting or blocking tlr7 in mice, resulted in protection from carcinogenic progression [77].

The increase in reactive oxygen species (ROS) is often observed in the progression of apoptosis or necrosis induced by cytotoxic drugs. The release of ROS and a shift in tumor cell metabolism can endow stromal fibroblasts, especially myofibroblasts, with a tumor-promoting phenotype [78]. For instance, fibroblasts acquire higher migratory capability through accumulation of hypoxia inducible factor 1α (HIF1α) and CXC-chemokine ligand 12 (CXCL12) when exposed to ROS [79].

Cancer cells releasing EVs upon apoptosis potentially have tumor-promoting effects. A study in gastric cancer showed that CAF induced apoptosis in the cancer cells, which resulted in the release of apoptotic vesicles from the cancer cells, educating the CAF to facilitate migration of cancer cells. This suggests CAF-mediated cancer cell apoptosis may promote cancer dissemination [80].

These studies are just a few examples of how conventional cytotoxic therapies potentially educate CAF to adopt a phenotype that can drive tumorigenesis (Figure 1). However, much is still unknown about the secondary effect of cell death in the TME with regards to the stromal population, and hence further studies are needed. Moreover, these studies highlight the importance of taking stromal cells into consideration when introducing cytotoxic therapies to the TME.

## 4. The Direct Effect of Cytotoxic Therapies on CAF

Cytotoxic therapies are the standard treatment in advanced PDAC, with the majority of patients not eligible for curative surgical resection. With CAF being the main cellular component, driving tumorigenesis, one could hypothesize that cytotoxic therapies could be used to eradicate CAF together with the tumor cells. Indeed, using standard chemotherapy regimens such as gemcitabine and nab-paclitaxel have resulted in a reduction of α-SMA-positive cells [81].

In comparison to cancer cells, CAF are considered to be relatively genetically stable, However, when fibroblasts are exposed to cytotoxic drugs or radiation-induced DNA damage, genetic mutations can be induced, resulting in CAF expressing a phenotype that contributes to the process of malignant transformation and further tumor progression [82,83]. There are studies supporting the concept that chemotherapy can phenotypically and metabolically change stromal fibroblasts into cancer-associated fibroblasts, leading to the emergence of a highly glycolytic, catabolic and inflammatory TME, subsequently contributing to tumor progression [78]. For example, stromal cells when stressed, secrete lactate. Lactate by itself can be used as mitochondrial fuel by cancer cells. Moreover, lactate secretion by stromal cells results in an acidic microenvironment, which has been shown to contribute to chemotherapy resistance and enhanced metastasis [84,85,86].

This idea of chemotherapeutics activating the tumor stroma, resulting in a more aggressive phenotype, has been suggested in several cancer models. Chemotherapy can stimulate HIF1, NFkB, SMAD, STAT3 and JNK/AP1 stress-linked signaling pathways in CAF, leading to a microenvironment which in turn can promote cancer proliferation and educate immune cells to an immunosuppressive phenotype, making cancer more invasive [78,87,88]. Stromal cells in prostate cancer exposed to the topoisomerase II inhibitor Mitoxantrone, a genotoxic agent, secreted Wnt16B, which in turn promoted the proliferation and invasion of cancer cells [89], and in colorectal cancer, FOLFOX chemotherapy exposure led to the enrichment of IL-17A–producing CAF, resulting in enhanced tumor growth and proliferation of cancer-initiating cells [90]. CAF treated with multiple chemotherapeutic agents in co-culture with breast cancer cells have also been shown to activate Hh signaling [78]. The three Hh genes identified in mice and in humans are sonic hedgehog (*Shh*), desert hedgehog (*Dhh*), and Indian hedgehog (*Ihh*), all involved in conserved pathways including cell differentiation, proliferation and survival [91]. Deregulation of the Hh is associated with the formation of many types of cancer [92], and stromal cell Hh production has been linked to cancer cell proliferation and metastasis [93]. 

In addition to apoptosis, cell senescence can be induced by chemotherapy in those cells sensitive to the cytotoxic agent. Senescence results in proliferation arrest. However, as a consequence of cell transformation it has been observed that clones can emerge that restart proliferation and become more aggressive [94]. In different cancer types, senescence induction in CAF has been associated with fueling oncogenic processes by enhancing proliferation of tumor cells in different cancers [95,96]. When CAF in desmoplastic tumors were treated with the maximum tolerated dose of chemotherapy, CAF underwent senescence and resulted in an activation of NFkB and STAT1, leading to the secretion of chemokines through the chemokine receptor CXCR1/2 axis [97]. The secreted cytokines triggered the phenotypic conversion of cancer cells into stem-like tumor-initiating cells, promoted tumor cell invasion, angiogenesis and the recruitment of myeloid-derived suppressor cells (MDSCs) [97]. Apart from inducing NFkB and STAT signaling, senescent cells have been shown to release paracrine factors including IL-1β, IL-6, CXCL8 and VEGF which are all cytokines known to be involved in tumor progression. For instance, CXCL8, shown to be upregulated in senescent CAF, signals via the CXCR1/2 axis. In PDAC, the CXCR2 axis is involved in MDSC recruitment, angiogenesis, tumor cell proliferation and migration. Further, upregulation of the CXCR2-axis in PDAC is associated with tumor-supporting inflammation, immunosuppression, angiogenesis and tumor growth [98,99,100].

Studies have shown that CAF exposed to chemotherapy release EVs that in turn can educate surrounding cells to a pro-oncogenic state and enhance treatment resistance in cancer cells. MSCs under stress increase osteosarcoma migration and apoptosis resistance via the release of EVs containing microRNAs associated with metabolic and metastasis-related genes [101]. CAF exposed to gemcitabine led to EVs being released from the CAF which in turn were taken up by tumor cells increasing Snail expression, which is associated with chemotherapy resistance, promoting proliferation and drug resistance. However, when subjecting the gemcitabine-exposed CAF to an EV-releasing inhibitor in co-culture conditions with pancreatic cancer cells, the survival of the cancer cells was reduced [102], demonstrating the role of EVs in cancer cell survival.

Most cancer patients will under their course of treatment have radiotherapy. Even though the cancer cells are being targeted, ultimately, the surrounding TME, including the CAF will also be irradiated. CAF, however, will undergo senescence and permanent DNA damage rather than die when irradiated. Radiotherapy-treated CAF are activated and educated towards an unfavorable phenotype, leading to an altered secretome of factors associated with driving cancer cells to EMT, increased invasiveness and therapy resistance [103].

Even though PDAC is a relatively radiotherapy resistant cancer, it is a treatment option sometimes used in combination with chemotherapy. Very little is known about the fate of PDAC CAF when exposed to radiotherapy. Recently, it was shown that radiotherapy increased inducible nitric oxide synthase (iNOS) expression and nitric oxide (NO) secretion by CAF following radiotherapy which in turn increased iNOS/NO signaling in tumor cells through NFkB. Tumor cells educated by CAF subjected to radiotherapy increased the release of inflammatory cytokines; upon inhibition of iNOS when combined with radiotherapy, PDAC tumor growth was delayed in a mouse model [104].

Apoptotic cells are known to release PGE_2_, which can also be applied to apoptotic stromal cells. CAF-mediated PGE_2_ secretion is known to induce cancer cell invasion [105]. Stromal cell-derived PGE_2_ could have many pro-tumorigenic functions in the TME. For instance, stromal cells undergoing apoptosis reprogramed monocytes/macrophages after phagocytosis to upregulate COX2, IDO, PD-L1, IL-10 and release PGE_2_, resulting in inhibition of T-cell proliferation [106].

Together, these studies imply that conventional anti-cancer treatments can educate CAF to drive tumorigenesis (Figure 2) and suggest that there remain significant gaps in knowledge about the effect of systemic therapies on non-cancerous cells in the TME. Hence, more studies are needed to gain further insight into this area.

## 5. Depletion of CAF in the PDAC TME

Based upon the growing body of studies showing the role of CAF in tumor progression, there has been an extensive effort in targeting CAF in the PDAC stroma by selective depletion. Pre-clinical studies carried out in various cancer types including PDAC have shown promising results. For example, the conditional ablation of FAP^+^ cells using diphtheria toxin depletion in a PDAC model reduced tumor growth and re-established tissue homeostasis [107]. Further, genetic depletion of FAP-expressing cells in a subcutaneous model of PDAC permitted immunological control of growth [108]. Other encouraging FAP targeting strategies using vaccines [109], FAP antibody-drug conjugates [110], peptide targeted radionuclide therapy [111] and CAR-T cells directed at FAP^+^ cells [112] have also been reported. Early phase trial data evaluating the FAP-specific humanized monoclonal antibody confirmed the safety and tolerability of the agent but with minimal evidence of tumor response. A phase I trial evaluating RO6874281, an FAP targeted IL-2 variant, has shown preliminary evidence of response in melanoma and squamous cell cancers, and trials are currently ongoing including in combination with PD-L1 inhibitors and other drug classes [113]. These data leave the question of efficacy somewhat unanswered, as stromal biology differs in different cancers. Other challenges in targeting FAP remain, for example, FAP is not solely expressed in CAF; hence, there are potential risks of systemic toxicity.

Even though some studies have been encouraging with regards to targeting stromal cells, there are also numerous studies attempting to target CAF that have yielded unexpected results reflecting the heterogeneity of CAF and questioning whether CAF only have tumor-promoting functions [114]. Several studies in multiple cancer types, including PDAC, have shown correlations between the number of α-SMA^+^ CAF and poor clinical outcomes [115,116,117,118]; however in a murine model when α-SMA-expressing cells were genetically depleted in pancreatic cancer, poorly differentiated tumors and shortened survival time were observed. Moreover, an increase regulatory T cells (Treg) infiltration was seen alongside a reduced number of F4/80^+^ macrophages [119,120].

Paracrine Hedgehog (Hh) signaling from cancer cells is known to activate CAF and promote stromal desmoplasia. Hence, blocking Hh signaling was hypothesized to lead to reduced fibrosis and desmoplastic formation in PDAC. When Hh-dependent CAF in a PDAC mouse model were genetically depleted, accumulation of an Hh-dependent stroma in all the stages of PDAC development was blocked. The mice with the Hh-depleted stroma developed tumors much faster than controls and their tumors were poorly differentiated with a higher tendency to metastasize [114]. Another approach to target Hh signaling is the inhibition of the transmembrane protein Smoothened (*SMO*), which prevents the induction of GLI transcriptional activity when cancer cells are exposed to Shh ligands. In a study where mice were treated with the SMO inhibitor IPI-926 prior to tumor formation, smaller tumors were seen compared to the controls; however, the tumors were more poorly differentiated than the controls and the mice had severe weight loss and died earlier [121], questioning whether targeting Hh-driven stroma could improve outcomes for PDAC patients. Further it has been shown that genetic or pharmacologic inhibition of Hh pathway activity accelerates progression of oncogenic Kras-driven disease in three different PDAC mouse models [122].

Even though there are a few preclinical studies with Hh inhibitors in combination with cytotoxic therapies that have shown some encouraging results [123,124], using Hh inhibitors in clinical trials has led to disappointing results, reflecting the above-mentioned animal studies [122,125]. For instance, using the Hh inhibitor saridegib in a phase II trial, median overall survival (OS) for the saridegib plus gemcitabine arm was less than six months which was less than the placebo arm, resulting in early trial discontinuation [126]. In addition, adding another Hh inhibitor, vismodegib, to gemcitabine in a Phase II trial failed to improve survival compared to gemcitabine alone in patients with metastatic PDAC [127]. Interestingly, a study evaluating Hh inhibition in PDAC demonstrated no statistically significant change from baseline epithelial (E-cadherin), stromal (α-SMA) or immune (CD45^+^) expression with either chemotherapy or vismodegib [125], highlighting the challenges associated with this approach.

Recently, it was discovered that Meflin, a membrane-bound glycosylphosphatidylinositol-anchored protein, is a marker for tumor-suppressive CAF in PDAC. Meflin was previously identified on MSCs in their undifferentiated state [22] and PSCs in the normal pancreas. The genetic ablation of Meflin in a PDAC mouse model led to poorly differentiated tumors. Further, Meflin deficiency resulted in straightened stromal collagen fibers, which is correlated with worse outcomes and more aggressive tumors in PDAC. Overexpression of Meflin in the mouse xenograft model resulted in suppression of tumor growth and upon analyzing 71 cases of human PDAC tissues, they found that infiltration of Meflin-positive CAF correlated with better patient outcomes [128], suggesting that Meflin-positive stromal cells could represent a population of tumor-suppressive CAF in PDAC.

We are now starting to document the heterogeneity of CAF, with some being shown to be tumor progressive and some tumor suppressive. As well as cancer cell-CAF crosstalk, with recent publications showing that different CAF populations can communicate with other stromal cells [129], and potentially each other, it is becoming increasingly important to consider these additional relationships. Further knowledge is needed to understand the mechanisms underlying the establishment of these phenotypes and the relationships between CAF and all cells within TME, in order to develop therapies that successfully ablate the most pathologic CAF population(s).

## 6. Targeting of CAF without Ablation or Apoptosis Induction

As discussed, depletion and/or direct cytotoxic targeting of the tumor-supportive stroma in PDAC may enhance tumor progression; hence, there is a need to explore other routes in targeting the tumor-supportive stroma. Induction of quiescence or reprograming (from tumor-promoting to tumor-suppressive phenotypes) may be more promising approaches. There have been studies suggesting that vitamin D analogues or derivatives (calcipotriol and all trans-retinoic acid) have the capability to reprogram activated PSCs to a more quiescent state, sensitize PDAC cells to chemotherapy and inhibit tumor progression in mice [130,131,132]. Furthermore, calcipotriol has also been shown to upregulate Meflin expression in PDAC CAF [128], supporting the idea that vitamin D analogues could possibly reprogram tumor progressive CAF to tumor suppressive CAF [133]. There are current ongoing clinical trials for PDAC patients using chemotherapy and immune checkpoint inhibitors in combination with vitamin D analogues which are showing promising results [134,135].

Minnelide is a synthetic and water-soluble pro-form of triptolide, a diterpenoid epoxide produced by the thunder god vine. When treated with Minnelide, CAF can be educated into a more quiescent state via TGF-β1 inhibition [136] which is accompanied by reduced collagen and HA deposition in PDAC tumors. Minnelide also improved functional vasculature and increased drug delivery to the PDAC stroma [137], which has also been seen when using verapamil [138]. These and other positive reports have led to Minnelide being used in several PDAC clinical trials including MinPAC, a phase II multicenter trial in patients with refractory disease [135].

Stromal expression of serum amyloid A3 (*SSA3*) in PDAC has shown to be associated with decreased survival and to be a pro-tumorigenic driver by the interaction of PDGFRα^+^ CAF and cancer cells [139]. Tumor growth was inhibited in mice where the CAF lacked the expression of serum amyloid protein, suggesting that the inhibition of serum amyloid A in CAF might be advantageous for PDAC patients.

Pancreatic CAF have also been reprogrammed towards quiescence via activation of the tumor-suppressor p53 by the Nutlin-3a derivative RG7112, resulting in a reduction of murine pancreatic desmoplasia [140].

Curcumin, a natural polyphenol in the *Curcuma longa* plant, has been shown to have the ability to reprogram CAF to a phenotype resulting in inhibition of tumor progression [141,142]. Low-dose curcumin has shown to decrease the expression of α-SMA and vimentin in CAF and reverse pancreatic cancer cell EMT. Further, curcumin educated CAF suppressed pancreatic cancer migration and produced significantly fewer and smaller lung tumors in mice compared to the control [143].

The Rho-associated protein kinase (ROCK) has been shown to be a driver in PDAC tumorigenesis via its involvement in ECM synthesis, stress-fiber assembly, cell contraction and promotion of unfavorable ECM remodeling via CAF MMP secretion [10,144,145]. A ROCK inhibitor (AT13148) reduced cellular contractility, motility and collagen matrix invasion in vitro and reduced subcutaneous tumor growth and local invasion in vivo [146]. A phase I study was carried out with AT13148 for the treatment of solid tumors but disappointingly it was not progressed owing to its narrow therapeutic index and pharmacokinetic profile [147]. However, another ROCK inhibitor, Fausudil, suppressed PSC activation, decreased tumor collagen deposition and led to improved survival in a PDAC mouse model [148]. Further, Fasudil inhibited pro-tumorigenic ECM remodeling in PDAC-bearing mice [145], which suggests that targeting ROCK in PDAC could be of particular benefit to patients exhibiting highly desmoplastic tumors.

Conversion of PDAC CAF to a myoblastic phenotype without inducing apoptosis has also been achieved with the bioactive lipid Lipoxin a4 (*LXA4*). *LXA4* inhibited TGF-β-mediated transformation in PSCs by blocking pSmad2/3 signaling, restricted crosstalk between CAF and PDAC cells and inhibited tumor growth, attributed to a significant reduction in fibrosis in a PDAC mouse model [149].

Targeting upstream signaling pathways essential to the change from being tumor-restraining fibroblasts to a tumor-supporting phenotype might be another option. Studies using inhibitors alone or in combination with cytotoxic therapies have been carried out. Examples of inhibitors used are the FAK inhibitor defactinib [150], fibroblast growth factor receptor (FGFR) inhibitor erdafinib [151], TGF-β inhibitors [152,153], drugs targeting IL-6 and JAK/STAT CAF activation [154] and antagonists against connective tissue growth factor (CTFG) [73].

Another approach is attempting to modulate CAF matrix deposition. For instance, PEGPH20, a pegylated hyaluronidase enzyme, has the ability to deplete hyaluronic acid produced by CAF. Hyaluronidase treatment showed promise in animal models and early clinical trials by improving progression-free survival in some PDAC patients [155,156]. Unfortunately, when PEGPH20 was tested in a phase III study, the development of the drug was stopped as it failed to improve overall survival in PDAC patients when used in combination with standard chemotherapy [157]. This study required patients to have hyaluronan-high tumors, defined as ≥50% hyaluronan staining in the ECM of tumor samples (analyzed centrally with a hyaluronan affinity histochemistry assay) and this highlights the potential pitfalls of identifying patients for stromal-targeting therapies based on tissue from metastatic versus primary sites of tumors [158].

These studies suggests that there are multiple strategies to explore when considering the targeting of CAF without fueling pro-oncogenic processes via direct apoptosis induction or depletion (Figure 3). This gives hope that targeting CAF in the PDAC stroma using more nuanced approaches could improve survival outcomes for PDAC patients.

## 7. Concluding Remarks

Despite being the main component of the TME in desmoplastic tumors such as PDAC, CAF were for a long time overlooked, especially when administering anti-cancer therapies. As discussed in this review, we now understand that inducing cell death with cytotoxic therapies can drive CAF-dependent tumor progression either directly or indirectly. Eliminating CAF from the PDAC TME is therefore a viable and evidence-led approach; however, many studies have highlighted the complexity of CAF, indicating that they could have both tumor supportive and suppressive roles. It is likely that future studies will look to target or modulate specific CAF phenotypes to impact pathology in a more controlled manner.

To target specific CAF phenotypes in PDAC, there are still many unanswered questions that need to be addressed. First, we need to improve our understanding of the origin of different CAF and identify more CAF-subpopulation markers for differentiating tumor-promoting versus tumor-suppressive CAF phenotypes. We then need a better understanding of how CAF heterogeneity varies in individual patients and cancer subtypes and whether we can stratify PDAC patients based on stromal composition to yield improved therapeutic outcomes.

By answering the above questions in addition to gaining better knowledge of how anti-cancer therapies directly and indirectly educate CAF in the TME, hopefully strategies to target PDAC cells and its surrounding tumor-promoting stroma can be optimized, resulting in improved survival for patients.

## Figures and Tables

**Figure 1 cells-10-01653-f001:**
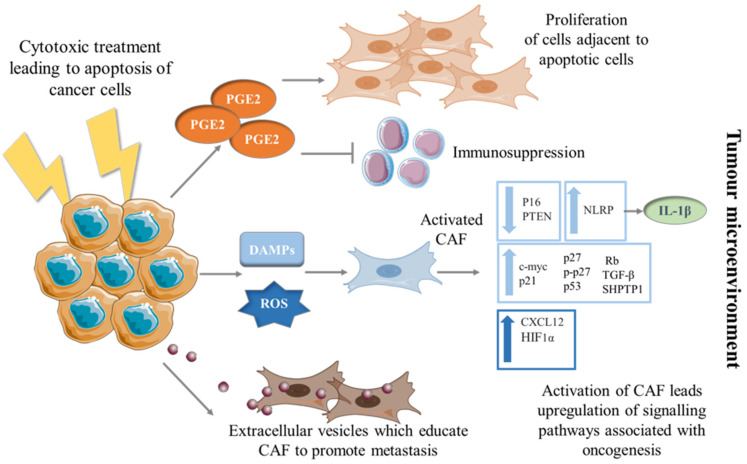
Education of CAF into a pro-tumorigenic phenotype by dying cancer cells. Mechanisms by which CAF can be transformed to a phenotype promoting tumorigenesis via the education of dying cancer cells. Cancer cells undergoing apoptosis secrete PGE2 which in turn can have pro-proliferative and immunosuppressive effects on surrounding cells in the TME. When cancer cells are exposed to cytotoxic therapies, they release DAMPs and ROS which stimulate CAF to differentiate into unfavorable phenotypes; upregulating pathways and secreting molecules associated with progression. Apoptotic cancer cells release EVs which have shown to be taken up by fibroblasts and educate them, resulting in CAF-promoting cancer cell metastasis.

**Figure 2 cells-10-01653-f002:**
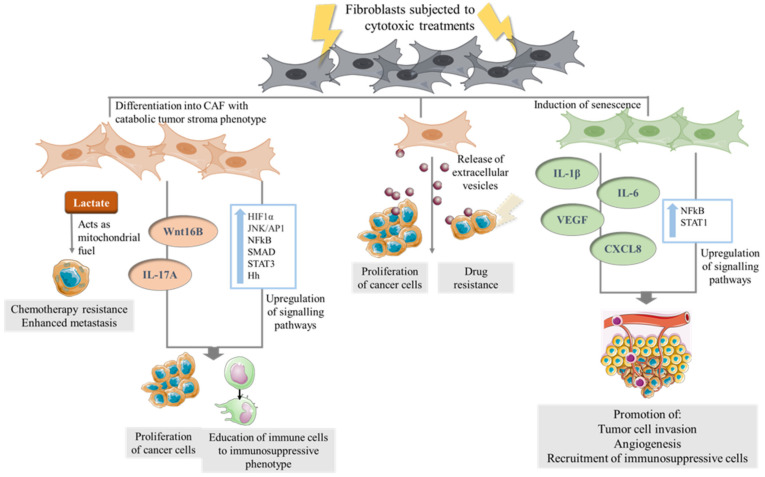
The direct effect of cytotoxic therapies on CAF. Fibroblasts in the TME of cancer cells which are subjected to cytotoxic therapies undergo transformation to a catabolic tumor stroma phenotype secreting metabolites such as lactate which in turn serves as mitochondrial fuel for cancer cells. Chemotherapy can stimulate stress-linked signaling pathways in CAF and the secretion cytokines lead to a microenvironment which in turn can promote cancer proliferation and educate immune cells to an immunosuppressive phenotype. Senescence induction in CAF have been shown to upregulate signaling pathways, leading to secretion of cytokines which promote tumor cell invasion, angiogenesis and the recruitment of myeloid-derived suppressor cells (MDSCs). CAF exposed to cytotoxic therapies release EVs which are taken up by cancer cells, leading to increased expression of genes associated with increased proliferation and drug resistance.

**Figure 3 cells-10-01653-f003:**
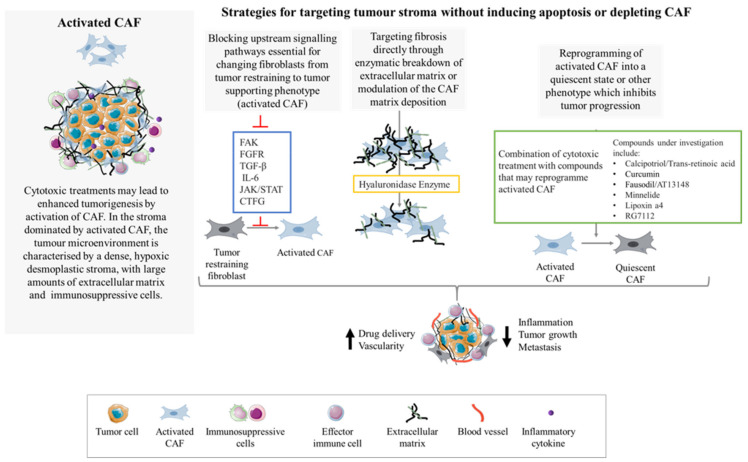
Targeting CAF without ablation or apoptosis induction. Activated CAF fuel desmoplasia, fibrosis, inflammation and tumor growth. To prevent or suppress the activation and formation of the desmoplastic reaction, strategies such as targeting upstream signaling pathways for CAF activation or reprogramming of activated CAF into a quiescent phenotype have been trialed. Another strategy is to target fibrosis directly with enzymatic breakdown of the ECM.

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
