# Peer review of "Apoptosis in the Pancreatic Cancer Tumor Microenvironment—The Double-Edged Sword of Cancer-Associated Fibroblasts"

_cells, 2021, doi:10.3390/cells10071653_

Round 1
Reviewer 1 Report
In this review article, Pfeifer et al. describe the different roles of carcinoma associated fibroblasts within the tumor microenvironment of pancreatic cancer. In this review the authors emphasize the unwanted effects of current therapeutic approaches that enhance CAF activity further fueling cancer progression. Overall this review s well written and discuss the potential molecular mechanisms responsible for these effects.
There are a few questions:
- Originally the word CAF (singular) has been used to describe a mixed population of cells. The plural version CAF (CAFs) used in this manuscript may sound redundant.
- The authors elegantly present the deleterious effects of inducing apoptosis with conventional therapeutic approaches on CAF biology. Similar to cancer cell heterogeneity, tumor resistant could also be driven by CAF heterogeneity. In addition to cancer cell-CAF crosstalk, recent literature suggest communication between CAF populations. Our poor understanding of fibroblast heterogeneity and their intercommunication in benign and malignant state might be responsible for the lack of success targeting CAF. Perhaps this aspect need to be emphasized in section 5. A discussion of targeting not just one but multiple pro-tumorigenic CAF subpopulations and their potential benefits.
- Page 2, line 87 the word “mentioned” is misspelled.
- Page 5, line 216 the verb “to be” is used in the present and past tense, please correct it
- Page 6 line 261, do the authors mean “..to eradicate CAFs together with tumor cells..”?, instead of “irradicate”?
- Font size in figure 3 are too small
Author Response
In this review article, Pfeifer et al. describe the different roles of carcinoma associated fibroblasts within the tumor microenvironment of pancreatic cancer. In this review the authors emphasize the unwanted effects of current therapeutic approaches that enhance CAF activity further fueling cancer progression. Overall this review is well written and discuss the potential molecular mechanisms responsible for these effects.
Many thanks to the reviewer for their kind comments.
There are a few questions:
Originally the word CAF (singular) has been used to describe a mixed population of cells. The plural version CAF (CAFs) used in this manuscript may sound redundant.
Many thanks for picking this up; we have changed CAFs to CAF throughout the manuscript.
The authors elegantly present the deleterious effects of inducing apoptosis with conventional therapeutic approaches on CAF biology. Similar to cancer cell heterogeneity, tumor resistant could also be driven by CAF heterogeneity. In addition to cancer cell-CAF crosstalk, recent literature suggest communication between CAF populations. Our poor understanding of fibroblast heterogeneity and their intercommunication in benign and malignant state might be responsible for the lack of success targeting CAF. Perhaps this aspect need to be emphasized in section 5. A discussion of targeting not just one but multiple pro-tumorigenic CAF subpopulations and their potential benefits.
We agree that CAF’s crosstalk with other cells, including other CAF, should be mentioned in section 5 and we have altered the final paragraph to reflect this (below). We are also very much on the same page as the reviewer in that, ultimately, multiple CAF populations may need to be targeted, once we understand their specific roles – this point has also been included. Many thanks.
“We are now starting to document the heterogeneity of CAF, with some being shown to be tumor progressive and some tumor suppressive. As well as cancer cell-CAF crosstalk, with recent publications showing that different CAF populations can communicate with other stromal cells, and potentially each other, it is becoming increasingly important to consider these additional relationships. Further knowledge is needed to understand the mechanisms underlying the establishment of these phenotypes and the relationships between CAF and all cells within the TME, in order to move towards therapies that successfully ablate the most pathologic population(s).”
Page 2, line 87 the word “mentioned” is misspelled.
Apologies for missing this – corrected.
Page 5, line 216 the verb “to be” is used in the present and past tense, please correct it
Many thanks; corrected.
Page 6 line 261, do the authors mean “..to eradicate CAFs together with tumor cells..”?, instead of “irradicate”?
Apologies; corrected.
Font size in figure 3 are too small
Font size in figure 3 has been increased. Many thanks.
Reviewer 2 Report
This article is about to discuss an important field in cancer biology and therapeutics. After a careful reviewing, this reviewer would like to raise the following comments:
(1). This review article focuses on therapy-induced desmoplasia and the underlying cancer-associated fibroblasts (CAFs). But actually, a highly desmoplastic stroma has already existed to constitute PDAC tissue when the disease can be diagnosed. In that tumor microenvironment, CAFs per se have produced abundant proteins such as PGE2 and TGF-beta to interplay with other types of cells including cancer cells and immune cells.
(2). It is known that 30-40% of CAFs in PDAC can be derived from the endothelial-mesenchymal transition (EndoMT) of endothelial cells. These EndoMT-derived cells have been further suggested to be correlated with PDAC malignancy by promoting macrophage M2 polarization and tumor growth (J. Hematol. Oncol. 12: 138, 2019). However, this review article cited some mesothelial cell papers but neglected the potential resource from endothelial cells (page 2, lines 62-72).
(3). It is generally thought that apoptotic tumor cells can be cleared by M2-type macrophages via efferocytosis to reduce tissue inflammation and thus be advantageous to tumor growth. Besides efferocytosis, M2-type macrophages secrete Th2-cytokines like IL-10 and TGF-beta to polarize Th1 T cells toward Th2. Therefore, the text “Efferocytosis can also a) polarise tumour-associated macrophages (TAMs) towards a pro-tumorigenic M2-like wound- healing phenotype, and b) recruit Tregs, further suppressing effector T cells.” (page 4, lines 155-157) is not a complete and clear statement.
(4). Four wording errors: “extracellular” (page 1, line 42 and labels in Figures 1 & 2) and “invasion” (page 8, line 335).
Author Response
This article is about to discuss an important field in cancer biology and therapeutics. After a careful reviewing, this reviewer would like to raise the following comments:
(1). This review article focuses on therapy-induced desmoplasia and the underlying cancer-associated fibroblasts (CAFs). But actually, a highly desmoplastic stroma has already existed to constitute PDAC tissue when the disease can be diagnosed. In that tumor microenvironment, CAFs per se have produced abundant proteins such as PGE2 and TGF-beta to interplay with other types of cells including cancer cells and immune cells.
We agree that there could be confusion regarding the status of ‘baseline / pre-treatment’ CAF versus those which have been affected, directly or indirectly, by treatment. We have adjusted the paragraph below, within the introduction to make it clearer that a) CAF are present at diagnosis and b) they are producing the factors mentioned.
“PDAC is largely resistant to conventional drug therapy and this is partially attributed to its complex and heterogenous tumor microenvironment (TME), with typically about 85% of the tumor being composed of a very dense, hypoxic, desmoplastic stroma depleted of immune effector cells upon diganosis5. Cancer associated fibroblasts (CAF) are the main cellular component in PDAC and once activated by injury or chronic inflammation they deposit large amounts of extracellular matrix (ECM), including laminins, fibronectins, collagens, and hyaluronan, as well as other factors, such as TGFβ1 and PGE2.”
(2). It is known that 30-40% of CAFs in PDAC can be derived from the endothelial-mesenchymal transition (EndoMT) of endothelial cells. These EndoMT-derived cells have been further suggested to be correlated with PDAC malignancy by promoting macrophage M2 polarization and tumor growth (J. Hematol. Oncol. 12: 138, 2019). However, this review article cited some mesothelial cell papers but neglected the potential resource from endothelial cells (page 2, lines 62-72).
Many thanks for the excellent point. We have removed endothelial cells from alongside adipocytes and pericytes and placed them alongside epithelial cells, as below. We have also included the reference mentioned.
“Epithelial or endothelial cells within the pancreas can undergo epithelial or endothelial to mesenchymal transition (EMT or EndoMT) and exhibit a fibroblastic phenotype (move original ref and include theirs). Other minority groups of cells believed to be educated to CAF-like cells include adipocytes and pericytes19 20 21 22.”
(3). It is generally thought that apoptotic tumor cells can be cleared by M2-type macrophages via efferocytosis to reduce tissue inflammation and thus be advantageous to tumor growth. Besides efferocytosis, M2-type macrophages secrete Th2-cytokines like IL-10 and TGF-beta to polarize Th1 T cells toward Th2. Therefore, the text “Efferocytosis can also a) polarise tumour-associated macrophages (TAMs) towards a pro-tumorigenic M2-like wound- healing phenotype, and b) recruit Tregs, further suppressing effector T cells.” (page 4, lines 155-157) is not a complete and clear statement.
Many thanks for the comment. We agree that this paragraph is not clear. We have reworked it (below) making it, in our view, clearer and more concise.
“Apoptotic cell clearance via efferocytosis polarizes tumor-associated macrophages (TAM) to adopt pro-tumor ‘M2’ wound-healing phenotypes which, by secreting cytokines such as IL‐4, IL‐10, IL‐13, and TGF‐β1, further educate Th0 or Th1 CD4 T cells towards a Th2 phenotype 42. Efferocytosis can also recruit Tregs, further suppressing effector T cells. These changes result in ineffective or severely compromised anti-cancer immune responses.”
(4). Four wording errors: “extracellular” (page 1, line 42 and labels in Figures 1 & 2) and “invasion” (page 8, line 335).
Many thanks for picking up these errors; now corrected.
Reviewer 3 Report
I have read the review entitled "Apoptosis in the pancreatic cancer tumour microenvironment - the double-edged sword of cancer associated fibroblasts." by Pfeifer and colleagues. This is a well written review.
Minor points should be addressed before publication:
Lines 334-339: Maybe it would be better to place this paragraph in the third section rather than the 4th which is more dedicated to treatments.
Line 436: It would be interesting to describe in a few words what type of therapeutic agent is the Minnelide and add an example of clinical trial that uses Minnelide (e.g. MinPAC).
Lines 448-453: It also been shown that curcumin had synergistic anti-tumour effects with gemcitabine or docetaxel on pancreatic cancer cells (https://doi.org/10.3892/or.2020.7713). It might be of interest to add this.
Resolution and quality of Figure 3 should be improved
Some abbreviations are missing (lines 227, 357, 388, 437,...)
Author Response
I have read the review entitled "Apoptosis in the pancreatic cancer tumour microenvironment - the double-edged sword of cancer associated fibroblasts." by Pfeifer and colleagues. This is a well written review.
Many thanks for the reviewer's kind comment.
Minor points should be addressed before publication:
Lines 334-339: Maybe it would be better to place this paragraph in the third section rather than the 4th which is more dedicated to treatments.
Many thanks for the comment. After considered discussion, we feel this paragraph would be better left in its current location. This is because the 3rd section discusses apoptosis of the tumour cells and not the CAF, whereas the 4th section discusses direct apoptosis of the CAF.
Line 436: It would be interesting to describe in a few words what type of therapeutic agent is the Minnelide and add an example of clinical trial that uses Minnelide (e.g. MinPAC).
Many thanks for the suggestion, we very much agree and have included two more sentences to explain what Minnelide is and its use in clinical trials, including the MinPAC trial.
Lines 448-453: It also been shown that curcumin had synergistic anti-tumour effects with gemcitabine or docetaxel on pancreatic cancer cells (https://doi.org/10.3892/or.2020.7713). It might be of interest to add this.
Many thanks for suggestion however, having reviewed this particular paper, we feel its emphasis is on the effects of curcumin (+chemo) on PDAC cells themselves, without assessing effects on CAF, meaning it doesn’t easily fit within this particular review and its focus on CAF.
Resolution and quality of Figure 3 should be improved.
Many thanks for the correction. We have increased the font size and improved the resolution.
Some abbreviations are missing (lines 227, 357, 388, 437,...)
Many thanks for seeing these, they have been corrected.
Reviewer 4 Report
The current manuscript reviews the role of cytotoxic therapy in pancreatic cancer in relation to the induction of apoptosis and its effects on cancer-associated fibroblasts. Further, therapeutic options targeting cancer-associated fibroblasts are summarized.
This is an interesting and comprehensive review of an important topic. The relevant literature is cited and discussed. The article is well organized and nicely illustrated. I have no further comments. Congratulations on an interesting piece of work.
Author Response
The current manuscript reviews the role of cytotoxic therapy in pancreatic cancer in relation to the induction of apoptosis and its effects on cancer-associated fibroblasts. Further, therapeutic options targeting cancer-associated fibroblasts are summarized.
This is an interesting and comprehensive review of an important topic. The relevant literature is cited and discussed. The article is well organized and nicely illustrated. I have no further comments. Congratulations on an interesting piece of work.
Many thanks to the reviewer for their very kind comments.